# Survival Pathways of HIF-Deficient Tumour Cells: TCA Inhibition, Peroxisomal Fatty Acid Oxidation Activation and an AMPK-PGC-1α Hypoxia Sensor

**DOI:** 10.3390/cells11223595

**Published:** 2022-11-14

**Authors:** Monika A. Golinska, Marion Stubbs, Adrian L. Harris, Laszlo G. Boros, Madhu Basetti, Dominick J. O. McIntyre, John R. Griffiths

**Affiliations:** 1Cancer Research UK Cambridge Institute, Cambridge University, Li Ka Shing Centre, Cambridge CB2 0RE, UK; 2Hypoxia and Angiogenesis Group, Weatherall Institute of Molecular Medicine, Department of Oncology, University of Oxford, Oxford OX3 9DS, UK; 3Department of Pediatrics, Harbor-UCLA Medical Center, University of California Los Angeles School of Medicine, Los Angeles, CA 90502, USA; 4SiDMAP, LLC, and the Deutenomics Science Institute, 2990 S. Sepulveda BLVD. #300B, Culver City, CA 90064, USA; 5The Lundquist Institute for Biomedical Innovation at the Harbor-UCLA Medical Center, 1124 W Carson St, Torrance, CA 90502, USA; 6Submolecular Medical Sciences, Vrije University of Amsterdam, 1081 HV Amsterdam, The Netherlands

**Keywords:** HIF-1β deficiency, Hepa-1 c4 cells, hypoxia response, 1,2-^13^C_2_-labelled glucose, TCA, fatty acid oxidation, AMP-activated kinase, phospho-p38 MAPK, PPARα, PGC-1α

## Abstract

The HIF-1 and HIF-2 (HIF1/2) hypoxia responses are frequently upregulated in cancers, and HIF1/2 inhibitors are being developed as anticancer drugs. How could cancers resist anti-HIF1/2 therapy? We studied metabolic and molecular adaptations of HIF-1β-deficient Hepa-1c4, a hepatoma model lacking HIF1/2 signalling, which mimics a cancer treated by a totally effective anti-HIF1/2 agent. [1,2-^13^C_2_]-D-glucose metabolism was measured by SiDMAP metabolic profiling, gene expression by TaqMan, and metabolite concentrations by ^1^H MRS. HIF-1β-deficient Hepa-1c4 responded to hypoxia by increasing glucose uptake and lactate production. They showed higher glutamate, pyruvate dehydrogenase, citrate shuttle, and malonyl-CoA fluxes than normal Hepa-1 cells, whereas pyruvate carboxylase, TCA, and anaplerotic fluxes decreased. Hypoxic HIF-1β-deficient Hepa-1c4 cells increased expression of PGC-1α, phospho-p38 MAPK, and PPARα, suggesting AMPK pathway activation to survive hypoxia. They had higher intracellular acetate, and secreted more H_2_O_2_, suggesting increased peroxisomal fatty acid β-oxidation. Simultaneously increased fatty acid synthesis and degradation would have “wasted” ATP in Hepa-1c4 cells, thus raising the [AMP]:[ATP] ratio, and further contributing to the upregulation of the AMPK pathway. Since these tumour cells can proliferate without the HIF-1/2 pathways, combinations of HIF1/2 inhibitors with PGC-1α or AMPK inhibitors should be explored.

## 1. Introduction

The HIF-1 and HIF-2 (HIF1/2) pathways enable cells to respond to hypoxia by regulating the expression of numerous genes, including those associated with growth, glucose metabolism and angiogenesis [1]. Hypoxia is common in tumours because of their compromised blood supply, so HIF activation is often observed; some tumours such as clear-cell renal cancer also show constitutive activation of HIF-1 and HIF-2, which then act as oncogenes [2,3]. HIF-1, and more successfully HIF-2, are currently targets for anticancer drug development [4,5,6], and HIF-2 inhibitors have been tested in clinical trials and are now approved for therapy of tumours with constitutive HIF-2 activation from vHL mutation [7,8]. A combination of HIF-1 and HIF-2 inhibitors has also been proposed, both for synergistic effects and for personalization of therapy [9]. If it turns out that these novel agents effectively suppress HIF1/2 activity in the clinic, it seems likely that tumours will develop resistance mechanisms, so it will be important to predict what these mechanisms are likely to be and how they might be overcome [10]. It has been reported, for instance, that although blocking HIF-1 in melanomas decreases glycolysis, increased glutaminolysis enables continued tumour growth [11]. Our overall aim was, therefore, to see what adaptations cells deficient in HIF-1/2 can utilise, in order to anticipate how cells treated with anti-HIF therapy might behave, and to suggest drug combinations for further testing.

A plausible model in which to address these questions is the well-established Hepa-1 c4 mouse hepatoma (Hepa-c4) [12], which has a point mutation in the HIF-1β/ARNT gene and therefore lacks expression of HIF-1β [13]. Since HIF-1β is an indispensable dimer partner for both HIF-1α and HIF-2α, no functioning HIF-1 or HIF-2 complexes are formed in Hepa-c4 cells [14,15], making them a good model for a tumour that has become resistant to either anti-HIF-1 or anti-HIF-2 agents, or to agents that simultaneously inhibit HIF-1 and HIF-2. Both c4 tumours and cultured c4 cells are fully viable, although they exhibit delayed initiation of growth compared with wild-type (Hepa-WT) Hepa-1 cells [16,17] and tumours [16,18].

HIF-1 activation upregulates glycolysis by enhancing the expression of glycolytic enzymes and glucose transporters, allowing cells (both normal and malignant) to switch to anaerobic glycolysis during hypoxia. HIF-1/2-deficient Hepa-c4 cells and tumours showed decreased expression of many glycolytic enzymes and glucose transporters. One would, therefore, expect decreased glycolysis in c4 tumours and cultured c4 cells, but we previously found similar glucose uptake and lactate production in c4 tumours compared with Hepa-WT tumours [18]. Two oncofactors that are thought to mediate the HIF-1-independent upregulation of glycolysis were either unchanged (c-myc) or downregulated (akt) in Hepa-c4 cells relative to Hepa-WT cells [18], so they could not have upregulated c4 cell glycolysis.

In our earlier work, we found lower ATP concentrations and equivalent rises in ADP and AMP in cultured Hepa-c4 cells and tumours in vivo [18,19] as compared to their Hepa-WT counterparts. The high AMP and low ATP concentrations in Hepa-c4 tumours suppressed the normal inhibition of phosphofructokinase-1 (PFK-1, the main control enzyme in the glycolytic pathway) by ATP, and allowed it to be allosterically activated by AMP [18], thus explaining the normal glucose uptake and glycolysis in Hepa-c4 cells under both normoxia and hypoxia.

In the present study, we monitored carbon flux from [1,2-^13^C_2_]-D-glucose through the metabolic pathways of cultured Hepa-c4 and Hepa-WT cells, both in normoxia and hypoxia, using SiDMAP metabolic profiling [20,21,22,23]. Glucose, the main substrate of solid tumours [24,25,26,27] must provide both energy and the anabolic precursors for generating biomass. We therefore evaluated metabolic fluxes via the tricarboxylic acid cycle (TCA), citrate shuttle, fatty acid utilisation, and pentose phosphate (PPP) pathways (Figure 1).

Another likely consequence of the high [AMP]:[ATP] ratio in Hepa-c4 cells would be the activation of AMP-activated protein kinase (AMPK), a metabolic control enzyme that initiates a pathway coordinating cellular proliferation with carbon source availability [28], and we have previously shown increased phospho-AMPK expression in Hepa-c4 tumours [18]. In the present study, we looked at several downstream targets of the AMPK signalling pathway: PGC-1α, p38 MAPK, and PPARα. PGC-1α has been suggested as an alternative hypoxia-sensing mechanism [29,30] and it might, therefore, be utilised by HIF-deficient cells and tumours.

## 2. Materials and Method

### 2.1. Cell Culture

Hepa-1 WT and Hepa-1 c4 cells (obtained from the American Tissue Culture Collection, Hepa-1c1c7 ATCC^®^ CRL-2026™ and c4 (B13NBii1), ATCC^®^ CRL-2717™) were grown in alpha MEM medium supplemented with 10% FBS and 50,000 units of Penicillin/Streptomycin (all reagents from Invitrogen) and regularly tested for mycoplasma.

For normoxic experiments, a standard 5% CO_2_ incubator was used. For hypoxic experiments, cells were incubated at 1% O_2_ and 5% CO_2_ in a hypoxia workstation (Ruskinn InvivO_2_ 500). Cells were seeded at 10^6^ in 10 cm^2^ dishes and cultured in either normoxia or hypoxia for 72 h (*n* = 3).

### 2.2. Metabolite and Enzyme Assays

Glucose and lactate: Cells were cultured in either normoxia or hypoxia for 48 h. Glucose (unlabelled) was assayed in the culture medium before and after 48 h’ culture (*n* = 7), using an Amplex Red Glucose kit (Invitrogen); lactate (*n* = 5) was assayed spectrophotometrically in the same samples using a lactate assay kit (Abcam). Both glucose uptake and lactate production were corrected for cell number.

Hydrogen peroxide (H_2_O_2_) was assayed in tissue culture media using a kit from Enzo Lifesciences according to the manufacturer’s instructions.

Acetate was measured by ^1^H MRS spectroscopy as described previously [19] and as detailed in Appendix A.

### 2.3. ^13^C tracer Studies

Cells were grown for 48 h in pyruvate-free medium containing 18 mM [1,2-^13^C_2_]-D-glucose (*n* = 4). The medium was then collected, cells were washed twice in phosphate-buffered saline, and cell pellets were harvested. Metabolite extractions and analyses were performed as described previously [31,32,33]. Please see Appendix A for the ^13^C enrichment patterns. Fluxes of the PPP pathway, TCA cycling, pyruvate dehydrogenase (PDH), citrate release, and malonyl CoA synthesis were inferred from ^13^C-labelled isotopomer changes.

### 2.4. TaqMan Gene Expression and ELISA Measurements

The expression of PGC-1α and PPARα genes was measured using RNA obtained from cell extracts (*n* = 7) with an RNAeasy extraction kit (Qiagen). An amount of 500 ng of RNA was reverse-transcribed to cDNA using a high-capacity cDNA reverse transcription kit (Applied Biosystems) in a thermal cycler at: 25 °C for 10 min, 37 °C for 120 min, and 85 °C for 5 min. The cDNA was then transcribed to mRNA in a real-time PCR reaction according to the TaqMan gene expression assays protocol (Applied Biosystems) and was assayed with Taqman probes Mm00440939_m1, Mm00482488_m1, and Mm00607939_s1, which were used to detect PPARα, PGC-1α, and β-actin, respectively. Results are presented in relative mRNA units standardised to β-actin.

Phospho-p38 MAPK (Thr180/Tyr182) was assayed by a PathScan^®^ Sandwich ELISA Kit #7946 according to the manufacturer’s instructions.

### 2.5. Data Analysis and Statistical Methods

Metabolite concentrations, rtPCR, and ELISA results for Hepa-WT and Hepa-c4 cells under normoxic and hypoxic conditions were compared using the unpaired two-tailed *t*-test. All data are shown as mean  ±  SEM unless otherwise stated. Significance is assigned for *p*-values < 0.05.

Mass spectroscopic analyses were carried out by three independent data downloads using background and natural ^13^C subtraction with Chemstation software (Agilent, Palo Alto, CA, USA). Single injections of 1 μL samples were applied at a 10:1 split ratio by the automatic sampler and accepted only if the standard sample deviation was less than 1% of the normalized peak intensity for each isotopomer. Statistical analysis was performed using Student’s t-test for unpaired samples. Significance is assigned for *p*-values < 0.05.

## 3. Results

### 3.1. Hepa-1 HIF-1β-Deficient Cells Are Capable of Growth in Hypoxia

During the first three days, in both normoxia and hypoxia, the Hepa-c4 cells grew more slowly than Hepa-WT cells (Figure 2a); this was consistent with the initial delay in Hepa-c4 tumour growth that we had previously reported [17,18]. At 48 h (Figure 2b), the difference in growth between Hepa-WT and Hepa-c4 cells was maximal; there were significantly more Hepa-WT than Hepa-c4 cells (*p* < 0.001) when cultured under normoxia, and significantly more Hepa-WT cells under normoxia than under hypoxia (*p* < 0.05). Between 48 and 72 h of culture, however, the growth rates of the Hepa-c4 and Hepa-WT cells were very similar.

### 3.2. Glucose Consumption and Lactate Production Increases in Response to Hypoxia in HIF-1β-Deficient Hepa-1 Cells

In preliminary studies using unlabelled glucose (*n* = 7, Figure 2c), both Hepa-c4 and Hepa-WT cells had similar glucose consumption rates in normoxia (17.6 ± 0.92 and 17.5 ± 0.67 µmol per million cells for Hepa-WT and Hepa-c4, respectively, *p* > 0.1) and the glucose uptake of both cell types increased significantly in hypoxia (to 23.1 ± 1.29 µmol per million cells for Hepa-WT, *p* < 0.004 and 28.5 ± 1.1 µmol per million cells for Hepa-c4, *p* < 0.001). Hypoxic Hepa-c4 cells consumed significantly more glucose than hypoxic Hepa-WT cells (*p* < 0.01). Total lactate production (*n* = 5, Figure 2d) under normoxia showed no difference between the Hepa-c4 (35.2 ± 0.8 μmol/million) and Hepa-WT (36.5 ± 3.9 μmol/million) cells, and both cell types showed a significant 2-fold increase (*p* < 0.001) in lactate produced under hypoxia (increasing to 66.4 ± 1.1 and 64.7 ± 4.43 μmol per million cells for Hepa-WT and Hepa-c4, respectively). These results confirm our previous observation [17] that in hypoxia, Hepa-c4 cells took up significantly more glucose than Hepa-WT cells but secreted the same amount of lactate.

### 3.3. Hepa-1 HIF-1β-Deficient Cells Downregulate Their TCA Cycle Flux, Shuttle Citrate out, and Channel Carbons toward Malonyl-CoA Synthesis

Hepa-1 cells showed significant changes in their TCA metabolism (Figure 3).

Anaplerotic replenishment of TCA intermediates was lower in Hepa-c4 cells, both via pyruvate carboxylase and via other pathways. Pyruvate carboxylase flux (Figure 3a) was significantly lower in Hepa-c4 compared to Hepa-WT cells, both in normoxia and hypoxia (*p* < 0.001 and *p* < 0.005, respectively); it did not change significantly in hypoxia in either cell type. Anaplerotic flux through other pathways (Figure 3b) was also lower in Hepa-c4 compared to Hepa-WT cells. In normoxia this difference was significant (*p* < 0.01) but in hypoxia it was not, probably due to the high variability among hypoxic Hepa-WT samples. Neither Hepa-WT nor Hepa-c4 cells showed a significant change in either of the two anaplerotic fluxes in hypoxia.

In normoxia, the TCA flux (Figure 3c) was significantly lower in Hepa-c4 than in Hepa-WT cells (*p* < 0.001), as expected because of the decreases in their pyruvate carboxylase (PC) and anaplerotic fluxes. It was significantly higher in hypoxia than in normoxia in the Hepa-c4 cells (*p* < 0.01), but there was no significant effect of hypoxia on the Hepa-WT cells.

PDH flux (Figure 3d) was significantly higher in Hepa-c4 than in Hepa-WT cells, both in normoxia and hypoxia (*p* < 0.001 for both conditions). This was consistent with HIF-1 activating pyruvate dehydrogenase kinase (PDK), an inhibitor of the PDH complex, in Hepa-WT cells [34,35]. Neither Hepa-WT nor Hepa-c4 cells showed any significant change in PDH flux in hypoxia. This was expected in Hepa-c4 cells, since the absence of HIF would prevent them activating PDK and thus PDH; however, it was not expected in HIF-competent Hepa-WT cells.

There was significantly more shuttling (instead of cycling) of citrate (Figure 3e) out of the TCA in Hepa-c4 compared to Hepa-WT cells, both in normoxia and hypoxia (*p* < 0.001). However, the citrate shuttle rate was slightly but significantly (*p* < 0.03) lower in hypoxic than normoxic Hepa-c4 cells. This decreased flux of citrate molecules from the TCA in hypoxic vs. normoxic Hepa-c4 cells would contribute towards the unexpectedly higher TCA flux in hypoxic vs. normoxic Hepa-c4 cells previously mentioned (Figure 3c).

The rate of malonyl-CoA synthesis (Figure 3f) was significantly higher in Hepa-c4 cells than in Hepa-WT cells in both normoxia and hypoxia (*p* < 0.001 for both conditions). Hepa-WT cells synthesized a similar amount of malonyl-CoA regardless of oxygen tension, whereas Hepa-c4 cells reduced malonyl-CoA production when made hypoxic (*p* < 0.001).

### 3.4. Hepa-1 HIF-1β-Deficient Cells Show Increased Signalling via AMPK-PPARα-PGC1α

We previously showed [18] that the low [ATP]:[AMP] ratio in Hepa-c4 cells that allosterically activates PFK-1 also induces phosphorylation (and thus activation) of AMPK [28]. We had also found increased expression of phospho-AMPK in Hepa-c4 tumours which, as predicted, grew slower than Hepa-WT tumours [18]. AMPK has recently been found to phosphorylate the catalytic subunit of PDH, facilitating TCA flux and promoting metastasis [36]. If the elevated AMPK had induced that effect in the Hepa-c4 cells, it would have contributed to the significantly higher PDH flux in Hepa-c4 vs. Hepa-WT cells that we observed, both in normoxia and hypoxia (Figure 3d).

PGC-1α, a known target of the AMPK pathway [37] that has been proposed as a component of an alternative oxygen response system [29,30], was upregulated in hypoxia in Hepa-c4 compared to Hepa-WT cells. Using real-time PCR, we found significantly lower PGC-1α in hypoxic compared to normoxic Hepa-WT cells (*p* < 0.001), whereas it was significantly increased in hypoxic compared to normoxic Hepa-c4 cells (*p* < 0.03). PGC-1α expression was also significantly higher in hypoxic Hepa-c4 than hypoxic Hepa-WT cells (*p* < 0.001, Figure 4a).

As well as inducing target gene expression, PGC-1α is modified post-transcriptionally by AMPK, and also by phospho-p38 stress-activated mitogen-activated protein kinase (p38α MAPK). ELISA experiments showed increased phospho-p38α MAPK (Thr180/Tyr182) expression in Hepa-c4 cells as compared to Hepa-WT cells in normoxia and hypoxia; this increase was significant for normoxic Hepa-c4 cells (*p* < 0.01) but not for hypoxic Hepa-c4 cells, possibly due to the high standard error of those samples (Figure 4b).

We also measured PPARα, the downstream target of PGC-1α, which controls peroxisomal fatty acid oxidation (PFAO) [38]. PPARα expression, assessed by real-time PCR (Figure 4c), was similar for Hepa-WT cells cultured in both conditions and in normoxic Hepa-c4 cells. However, in hypoxia, it increased almost three-fold in Hepa-c4 cells (*p* < 0.001 when compared to normoxic Hepa-c4 and to hypoxic Hepa-WT cells).

### 3.5. Hepa-1 c4 Cells Show Increases in Peroxisomal Markers and PPP

To quantify PFAO flux, we measured its products, acetate and H_2_O_2_ (Figure 5a,b, respectively).

Both intracellular acetate content and H_2_O_2_ production were increased in Hepa-c4 cells in hypoxia compared to normoxia. The acetate content (Figure 5a) increased in Hepa-c4 cells compared to Hepa-WT cells under both conditions and in Hepa-c4 cells was significantly higher under hypoxia than normoxia (*p* < 0.05). The H_2_O_2_ content (Figure 5b) of hypoxic Hepa-c4 cell culture media was more than 3-fold higher than in the media from normoxic Hepa-c4 cells (*n* = 5), but this difference was not statistically significant as there was a very large standard error in the hypoxic sample data, possibly due to the spontaneous degradation of the H_2_O_2_ in some of the samples before they were analysed.

Significant, albeit small, flux differences were seen in ribose synthesis via the oxidative and non-oxidative branches of the pentose phosphate pathway (PPP) in Hepa-1 cells. Labelled ribose production via the oxidative PPP branch (Figure 5c) was significantly higher in Hepa-c4 cells than in Hepa-WT in both conditions (*p* < 0.02). In the Hepa-WT cells, oxidative ribose production significantly decreased (*p* < 0.001) in response to hypoxia, whereas it remained unchanged in the Hepa-c4 cells. Fractional ribose production via the non-oxidative PPP branch (Figure 5d) was slightly lower in Hepa-c4 cells, but this difference was only significant (*p* < 0.05) in hypoxia.

## 4. Discussion

In this study, we aimed to determine what survival pathways cancer cells will revert to in order to withstand hypoxic stress when the functioning of the HIF pathway is interrupted. We looked at adaptations of HIF-1/2-deficient Hepa-c4 cells as the model for fully effective anti-HIF therapy (both HIF-1 and HIF-2 inhibited).

Despite their malfunctioning HIF pathway, HIF-1β-deficient hepatomas continued growing in hypoxia, and both WT and HIF-deficient cells upregulated their glucose uptake and lactate production. As we reported previously, HIF-1β-deficient Hepa-c4 tumours showed an increased AMP:ATP ratio [18], suggesting that HIF-1β deficiency may be contributing to this energy balance change.

Anaplerotic generation of oxaloacetate (and thus the replenishment of TCA intermediates) via flux through the pyruvate carboxylase (Figure 3a) and other pathways (Figure 3b) was slower in Hepa-c4 cells than Hepa-WT cells and was unchanged by hypoxia in either cell type. Although TCA flux (i.e., the rate of TCA cycling) in normoxia was significantly lower in Hepa-c4 than in Hepa-WT cells, it increased significantly when Hepa-c4 cells were made hypoxic, whereas Hepa-WT cells showed no significant effect of hypoxia (Figure 3c). This decreased TCA flux in Hepa-c4 cells was due to the exit of intermediates from the TCA before the first full turn: citrate shuttling to the cytoplasm was higher in Hepa-c4 cells (Figure 3e), and they converted more α-ketoglutarate to glutamate, which also exited the TCA before completion of the first TCA cycle turn (Figure 3c). Both effects would have reduced TCA cycling.

The increased PDH activity in Hepa-c4 cells (Figure 3d) could be due to the lower expression of PDK-1 (an inhibitor of PDH) observed in Hepa-c4 cells grown as tumours [18]. This increased PDH activity would not load the TCA, since the formation of acetyl-CoA is a cataplerotic reaction.

Citrate transported out of the mitochondrial membrane by the citrate shuttle can be broken down in the cytosol by citrate lyase to oxaloacetate and acetyl-CoA (Figure 3e). The latter will form malonyl-CoA, which can be used for de novo fatty acid synthesis, and a higher flux of that metabolite was indeed found (Figure 3f). Malonyl-CoA inhibits carnitine palmitoyltransferase I (CPT1), the enzyme that catalyses the transfer of fatty acids into mitochondria, which is the rate-limiting step in mitochondrial fatty acid β-oxidation [39]. Downregulation of the cpt1 gene is correlated with reduced fatty acid β-oxidation [40], so we postulate that increased malonyl-CoA in Hepa-c4 cells could channel fatty acids away from β-oxidation in the mitochondria. Instead, they would go to the peroxisomes, where the PFAO pathway metabolises long-chain saturated and unsaturated fatty acids by chain shortening, rather than complete oxidation to CO_2_ and water. These shorter-chain fatty acids are then transported to mitochondria by a CPT1-independent process (for a review, see [41]).

Since we observed that in HIF1/2-deficient Hepa-c4 cells: (1) malonyl-CoA flux was significantly increased (Figure 3f); (2) the TCA was downregulated (Figure 3c); and (3) the labelled acetyl-CoA pool was diluted with unlabelled fatty acid-derived acetyl-CoA (Figure 3a), we propose that the source of this unlabelled acetyl-CoA was unlabelled fatty acids oxidised in the PFAO pathway (illustrated in Figure 1). The by-products of PFAO are acetate and H_2_O_2_, both of which were increased in hypoxic Hepa-c4 cells compared to either hypoxic Hepa-WT cells or normoxic Hepa-c4 cells (Figure 5a,b).

Acetate is mainly generated by the PFAO, since mitochondria-derived acetate is negligible [42], so increased acetate in hypoxic Hepa-c4 cells supports our hypothesis. Moreover, the L-3-hydroxyacyl-CoA dehydrogenase reaction during PFAO requires NAD+ [43], which would most probably be supplied by peroxisomal oxidase converting oxygen to H_2_O_2_.

While it may seem unlikely that an oxidative pathway should be upregulated in cells cultured under hypoxia, a case for that can be made, bearing in mind that the hypoxic cells were cultured at 1% O_2_, so some oxidative metabolism would have been possible. First, the oxidative product of the PFAO pathway, H_2_O_2_ (present at elevated concentrations in hypoxic Hepa-c4 cells), is quickly broken down by the enzyme catalase (a constituent of peroxisomes) to O_2_ and water, thus in principle recycling the scarce O_2_. Second, the acetyl-CoA product of PFAO can be handled more easily in a hypoxic environment than can acetyl-CoA produced by mitochondrial β-oxidation, which would be oxidized via the TCA and the respiratory chain, thus demanding a substantial O_2_ supply. In contrast, acetyl-CoA formed during PFAO can be degraded by acyl-CoA hydrolase to acetate, another by-product that was elevated in hypoxic Hepa-c4 cells. The final PFAO by-product, NADH, could in principle be recycled to NAD+, either by conversion of pyruvate to lactate or by transporting its reducing equivalent into the mitochondria (via the malate-aspartate shuttle) followed by oxidation in the respiratory chain. The formation of lactate from pyruvate seems unlikely, since each pyruvate molecule generated by glycolysis already generates one NADH, which must be oxidized by formation of lactate to allow further glycolysis. Consequently, NADH oxidation in the mitochondria seems more likely, even though it would require some O_2_. The net effect of oxidizing fatty acids in the peroxisomes rather than the mitochondria would thus seem to be a reduction in O_2_ requirement approaching 50%; that could be a significant advantage to a hypoxic cell in which the energy metabolism had been disordered by the absence of HIF and inappropriate upregulation of the AMPK pathway.

H_2_O_2_ produced by the PFAO causes oxidative damage and must be scavenged by NADPH formed in the PPP. We found that oxidative PPP flux was significantly higher in Hepa-c4 than Hepa-WT cells (Figure 5c). Not only would that synthesize more ribose for nucleic acid metabolism, but the NADPH produced would provide a reducing equivalent for the synthesis of reduced glutathione. The latter neutralizes H_2_O_2_ by forming oxidised glutathione and water, thus protecting the cells against reactive oxygen species (ROS). As the hypoxic Hepa-c4 cells had higher levels of H_2_O_2_, a powerful intracellular oxidant, than Hepa-WT cells (Figure 5b), it is likely that they needed antioxidant protection. Interestingly, others have reported that increased steady-state H_2_O_2_ is associated with the metastatic phenotype in bladder tumour cell lines [44]. In addition, NADPH, coupled with the increased availability of acetyl-CoA associated with increased PDH flux (Figure 3d), may also have contributed to new lipid synthesis. Interestingly, the other product of PFAO—acetate, which we found increased in hypoxic Hepa-c4 cells, has also been recently suggested as an alternative carbon source that promotes lipid synthesis in cancer cells under hypoxia [45].

Malonyl CoA formation from acetyl CoA converts ATP to ADP, but PFAO breakdown of the resulting fatty acids does not resynthesise ATP, thus creating a futile cycle (see Figure 6).

Conversion of ADP to AMP by adenylate kinase lowers the ATP:AMP ratio, which explains our previous observations of PFK-1 activation and increased phospho-AMPK expression in Hepa-c4 tumours [18].

We looked further at expression of downstream targets of the AMPK signalling pathway: PGC-1α, p38 MAPK and PPARα, to elucidate the role of that pathway in the hypoxia response when HIF1/2 is not functional. All three proteins were higher in hypoxic Hepa-c4 cells (Figure 4a–c). Hypoxia, increased ROS concentrations, and exogenous treatment with H_2_O_2_ have all been shown to increase p38α MAPK expression [46]. Since H_2_O_2_ is a by-product of the PFAO, upregulation of p38α MAPK expression and peroxisomal activity (via increased H_2_O_2_ production) may be related.

AMPK also inhibits acetyl-CoA carboxylase, which synthesizes malonyl-CoA, but surprisingly, increased amounts of malonyl-CoA were found. However, increased malonyl-CoA levels have been observed without increased acetyl-CoA carboxylase activity [47], so we have postulated that despite the activation of the AMPK pathway, Hepa-c4 cells produce higher amounts of malonyl-CoA, which would then inhibit mitochondrial fatty acid β-oxidation and channel fatty acids to the peroxisomes. Interestingly, it has been proposed that de novo fatty acid synthesis is necessary for activation of PPARα and stimulation of fatty acid β-oxidation [48].

Peroxisomal β-oxidation is known to be PPARα-inducible; we found no difference in PPARα expression between normoxic Hepa-WT and Hepa-c4 cells or in hypoxic Hepa-WT cells, but it was upregulated in hypoxic Hepa-c4 cells (Figure 4c). In liver, PPARα maintains fatty acid homeostasis by stimulating fatty acid synthesis in response to increased fatty acid oxidation [48,49,50]. We hypothesize, therefore, that Hepa-c4 cells create a fatty acid synthesis/fatty acid oxidation cycle [48] to help them survive in the absence of HIF. The high malonyl CoA flux in Hepa-c4 HIF-deficient cells suggests that new lipid synthesis takes place, and either because the pO_2_ is too low or because a high malonyl CoA inhibits fatty-acyl CoA uptake into mitochondria, fatty acid oxidation is proceeding by the PFAO instead of mitochondrial β-oxidation. This apparently “futile” cycle would waste ATP, but this would benefit the Hepa-c4 cells by lowering the [ATP]:[AMP] ratio and thus upregulating both glycolysis and the PFAO (Figure 6). Interestingly, a HIF-1α binding site has been identified on the PPARα gene in intestinal epithelial cells, and PPARα downregulation was HIF-1-mediated [51]. Similarly, PPARα activation suppressed HIF-1 signalling in cancer cells [52]. In Hepa-c4 cells, the lack of functioning HIF-1/2 would increase PPARα expression.

Since AMPK is known to induce expression of PGC-1α [37], a proposed component of an alternative oxygen response system [29,30], we measured PGC-1α expression in response to hypoxia: it was downregulated in Hepa-WT cells and upregulated in Hepa-c4 cells (Figure 4a), suggesting that Hepa-c4 cells were indeed utilising that alternative oxygen response system. PGC-1α regulates oxidative and mitochondrial metabolism as well as the balance between fatty acid oxidation and synthesis [53,54]. PGC-1α induction by hypoxia has been shown in cells lacking ARNT, the aryl hydrocarbon receptor nuclear translocator, which is another function of HIF-1β [29], the absence of which is therefore the same defect as in Hepa-c4 cells. PGC-1α increases gluconeogenesis and glucose uptake [55,56,57], and although Hepa-c4 cells and tumours have decreased GLUT 1 and GLUT 3, we previously found similar expression of GLUT 2 (the liver-type glucose transporter) in both Hepa-c4 and Hepa-WT cells [18]. Thus, PGC-1α may increase glucose uptake in Hepa-c4 cells through GLUT 2.

As PPARα and its upstream cofactor PGC-1α were increased in hypoxic Hepa-c4 cells, and since both proteins have been linked to peroxisomal function and are major regulators of liver lipid metabolism and gluconeogenesis [58,59,60] (n.b. Hepa-1 cells are hepatoma cells, originally derived from hepatocytes), it is likely that PPARα and PGC-1α play important roles in the metabolic reprogramming of the Hepa-c4 cells and of tumours in general. These results suggest that the alternative hypoxia response mechanism involving PGC-1α would enable hypoxic Hepa-c4 cells to survive and grow in the complete absence of a functional HIF-1 or HIF-2 complex, as would be the case during perfectly successful anti-HIF cancer therapy.

## 5. Conclusions

We have shown that both hepatoma cell lines were able to grow in hypoxia despite the HIF-1β deficiency. This mimics the situation in a cancer that is resisting an anticancer drug that eliminates HIF-1/2; a similar approach was recently used to predict resistance mechanisms to lactate dehydrogenase inhibitors [61]. Both WT and HIF-1β-deficient cell lines cultured in hypoxia increased glucose uptake and lactate production suggesting that HIF1/2 was not required to upregulate glycolysis.

HIF1/2-deficient Hepa-c4 cells showed complex, and in some cases counter-intuitive, metabolic adaptations. The abnormally low ATP:AMP in Hepa-c4 cells maintains glycolysis but also activates AMPK, which upregulates pathways that are normally associated with low substrate availability. We propose that HIF-deficient Hepa-c4 tumour cells survive hypoxia by downregulating TCA cycle flux but diverting it to export citrate and by channelling fatty acids away from mitochondria for breakdown by the PFAO, thus producing high levels of acetate and H_2_O_2_. Additionally, elevated flux through the oxidative PPP branch helps protect them from the oxidative damage, sustains energy requirements, and maintains redox equilibrium.

Recently, a mechanism of maintaining the TCA cycle was reported [62] that utilised mitochondrially derived citrate exported to the cytoplasm, which was then metabolized by ATP citrate lyase. This produced acetyl-CoA and oxaloacetate, the latter being used to complete a non-canonical TCA cycle. In our work on HIF-deficient cancer in normoxic cells, we also found increased citrate exported to the cytoplasm. The acetyl-CoA then entered lipid metabolism. However, in our case the oxaloacetate was not utilised to maintain the Krebs cycle. Thus, there are several ways the citrate derived from the Krebs cycle is utilised to maintain metabolic fitness, depending on cell type and HIF function. This also highlights the normoxic role of HIF in Krebs cycle regulation.

We have also shown that three downstream targets of AMPK: PGC-1α, p38 MAPK, and PPARα, which may be involved in an alternate hypoxia response mechanism, are upregulated in Hepa-c4 cells. We propose that these metabolic adaptations (Figure 6) enable HIF-1β-deficient cells and tumours to survive without a functioning HIF-1 or HIF-2 pathway.

It remains to be elucidated what other factors might be involved in the hypoxic response; furthermore, AMPK, ERK, and JNK pathways are involved in the regulation of metabolic reprogramming in tumour cells, and it has been shown that hypoxia-driven phosphorylation of ERK1-2 influences TCA flux by activation of PDK-1 and the consequent inhibition of PDH [63]. Hypoxia-induced autophagy, together with elevated Ras/ERK signalling, has been reported to constitute a protective mechanism against low-oxygen tensions in cancer cells [64]. Interestingly, in mouse embryonic stem cells, chronic hypoxia caused the dephosphorylation of ERK independently of the HIF-1α expression [65]. Given the above, it would be interesting to examine whether ERK1-2 phosphorylation is differently affected in HIF-1β-deficient Hepa-1c4 cells under hypoxia.

The mechanisms that allow survival of hypoxic HIF-1/2-deficient tumour cells could explain the lack of success of anti-HIF therapy as a cancer treatment. It has been suggested that combination therapy involving drugs that inhibit HIF-1/2 activity should be considered [9]. The viability of HIF-1/2-deficient Hepa-c4 cells and tumours suggests that tumour proliferation will still be possible, even with both the HIF-1 and HIF-2 pathways suppressed, because alternative signalling and metabolic pathways can be activated. Future studies should test whether anti-HIF drugs do indeed induce these metabolic adaptations.

The present results are observed in a model that mimics a fully effective anti-HIF therapy. Deficiency in only HIF-1α would still enable the formation of HIF-2 and consequent signalling via this node, which may not trigger the aforementioned survival mechanisms. Further experiments are needed to see whether HIF-1β deficiency would evoke similar adaptations in other tissues.

Our results also suggest that suppression of HIF-1/2 will result in the activation of the p38 MAPK/PPARα/PGC-1α pathway. To date, most efforts have been targeted at the development of AMPK pathway activators and no specific and widely used AMPK inhibitors are available (as reviewed in [66]). However, as suggested in this article and shown by others [67], AMPK inhibition may constitute a new potential treatment strategy. For future therapies, combinations of anti-HIF-1/2 drugs with inhibitors of the PGC1α, PPARα, or AMPK pathways should, therefore, be considered.

## Figures and Tables

**Figure 1 cells-11-03595-f001:**
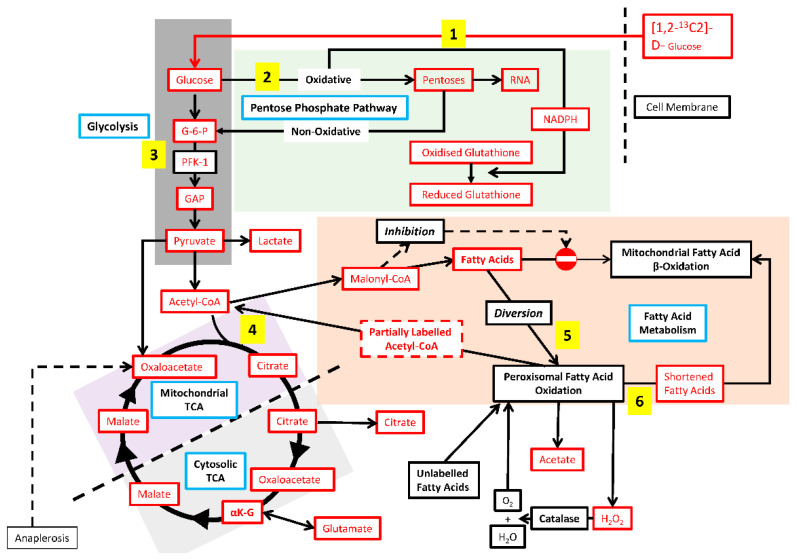
Diagram of metabolic pathways and relevant control mechanisms. 1—Glucose consumption, 2—Pentose phosphate pathway, 3—Rate-limiting step in glycolysis, 4—AcetylCoA to malonylCoA flux (de novo fatty acid synthesis), 5—Fatty acid diversion into peroxisomal fatty acid oxidation, 6—Shortened fatty acids enter mitochondria for further β-oxidation, 7—Citrate shuttle.

**Figure 2 cells-11-03595-f002:**
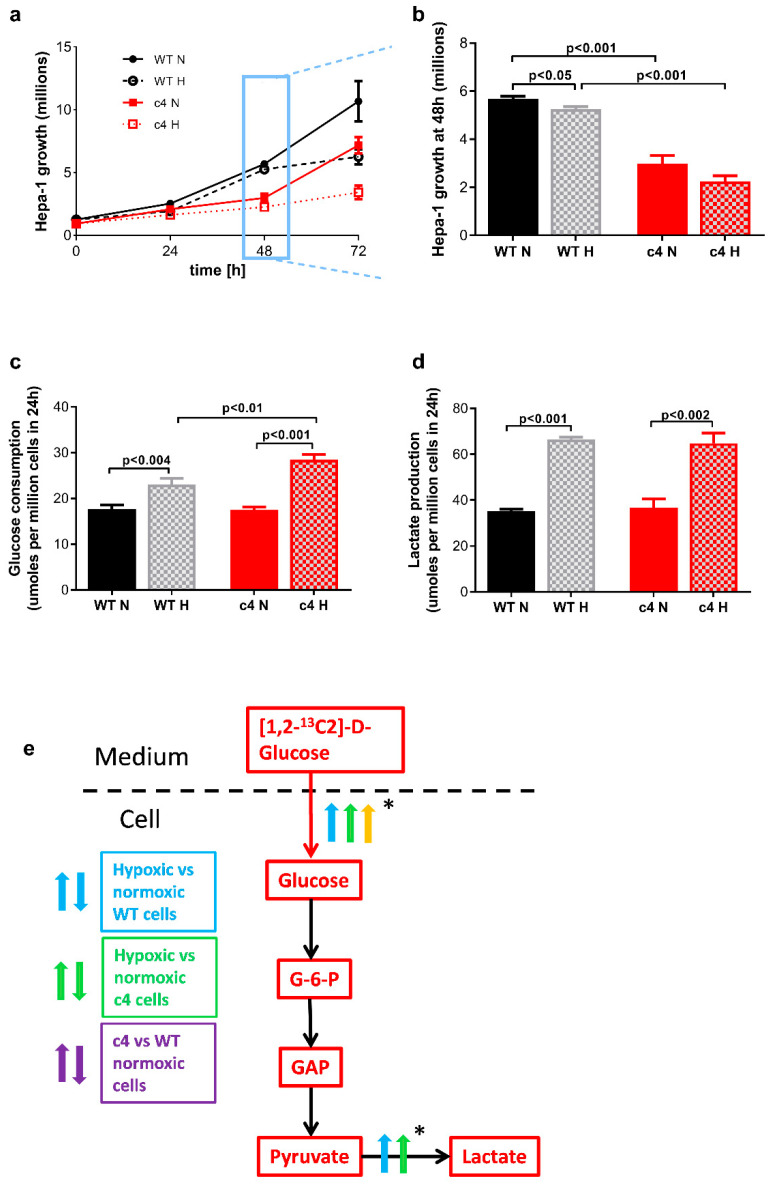
Cell growth, glucose consumption, and lactate output under normoxia and hypoxia in Hepa-1 WT and c4 cells. (**a**) Hepa-WT and Hepa-c4 growth profiles in normoxia and hypoxia (*n* = 3). (**b**) Cell numbers at 48 h showed delayed growth in Hepa-c4 cells (*n* = 3). (**c**) Glucose consumption at 48 h was elevated more by hypoxia in Hepa-c4 cells than in WT cells (*n* = 7). (**d**) Lactate production at 48 h was equally elevated by hypoxia in Hepa-c4 cells and WT cells (*n* = 5). (**e**) Diagram showing flux differences and changes between the two Hepa-1 cell types in normoxia and hypoxia. N, normoxia; H, hypoxia (1% O_2_); arrows pointing upwards suggest significant increase for a given comparison. * Indicates differences in metabolic flux (↑ increased flux; **↓** decreased flux) due to HIF deficiency and hypoxia, which were inferred from concentration changes in the substances indicated. Panels (**a**–**d**) show data expressed as mean ± SEM; statistical significance assessed by unpaired two-tailed *t*-test.

**Figure 3 cells-11-03595-f003:**
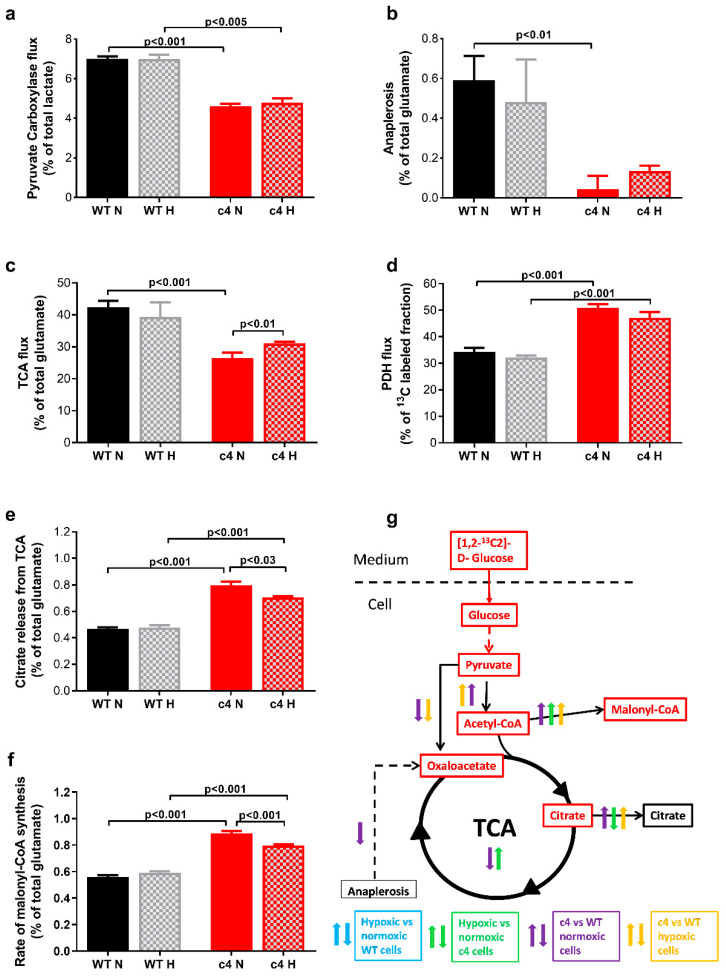
TCA cycle and related fluxes inferred from labelling patterns of ^13^C glutamate in Hepa-1 cells. HIF-deficient cells show: (**a**) reduced PC flux (oxaloacetate formation from pyruvate via pyruvate carboxylase; M4_m/z198). (**b**) reduced anaplerotic flux through the TCA cycle (ratio of pyruvate carboxylation to TCA cycle flux). (**c**) reduced TCA flux (fewer completed TCA cycles, M2, m2_m/z 152-m3_m/z 198). (**d**) increased PDH flux (M2, C2-C4 fragments, m/z 198) (**e**) increased release of citrate from the TCA cycle via the citrate shuttle (glutamate in equilibrium with citrate and α-ketoglutarate, m/z 198). (**f**) increased rate of new malonate synthesis and fatty acid production. (**g**) Diagram showing flux differences and changes between the two Hepa-1 cell types in normoxia and hypoxia. N, normoxia; H, hypoxia (1% O_2_), arrows pointing upwards suggest significant increase for a given comparison. Indicates differences in metabolic flux (↑ increased flux; **↓** decreased flux) due to HIF deficiency and hypoxia, which were inferred from changes in the relevant ^13^C-labelled isotopomers. Panels (**a**–**f**) show data expressed as mean ± SEM (*n* = 4, 3 technical replicates), statistical significance assessed by unpaired two-tailed *t*-test.

**Figure 4 cells-11-03595-f004:**
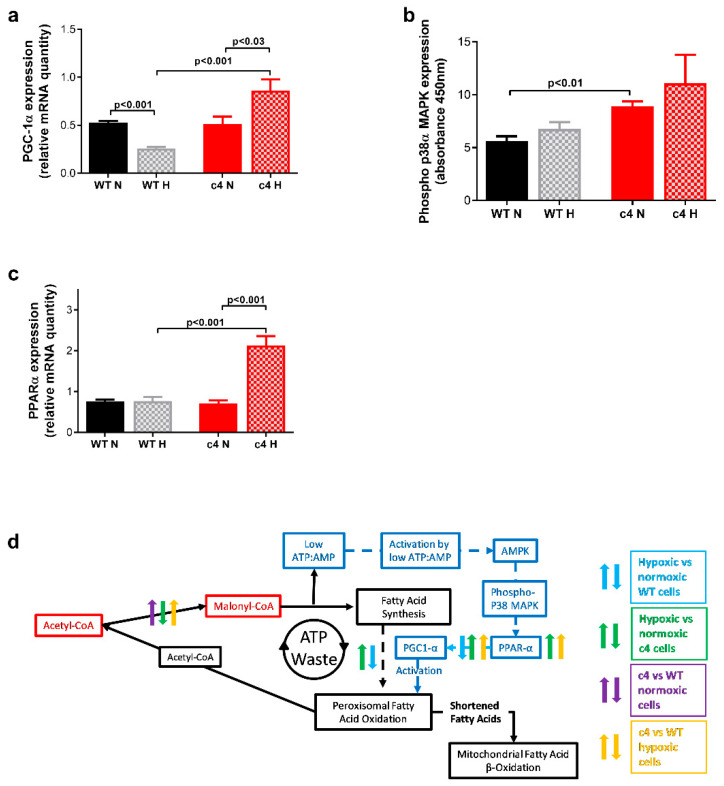
AMP/ATP-ratio-related signalling in Hepa-1 cells. (**a**) Hypoxia had opposing effects on PGC-1α expression in WT and Hepa-c4 cells; there was a two-fold increase in PGC-1α expression in hypoxic compared to normoxic Hepa-c4 cells (*p* < 0.001, *n* = 9), (**b**) Hepa-c4 cells in both normoxia and hypoxia showed increased expression of phospho-p38α MAPK (Thr180/Tyr182); this increase was significant for normoxic Hepa-c4 cells (*p* < 0.01, *n* = 3), (**c**) PPARα was increased by hypoxia in Hepa-c4 but not in Hepa-WT cells (*p* < 0.001, *n* = 9), (**d**) Diagram showing expression differences and changes between the two Hepa-1 cell types in normoxia and hypoxia. N, normoxia; H, hypoxia (1% O_2_); arrows pointing upwards suggest significant increase for a given comparison. Indicates differences in metabolic flux (↑ increased flux; **↓** decreased flux) due to HIF deficiency and hypoxia, which were inferred from changes in the relevant ^13^C-labelled isotopomers Panels (**a**–**d**) show data expressed as mean ± SEM; statistical significance assessed by unpaired two-tailed *t*-test.

**Figure 5 cells-11-03595-f005:**
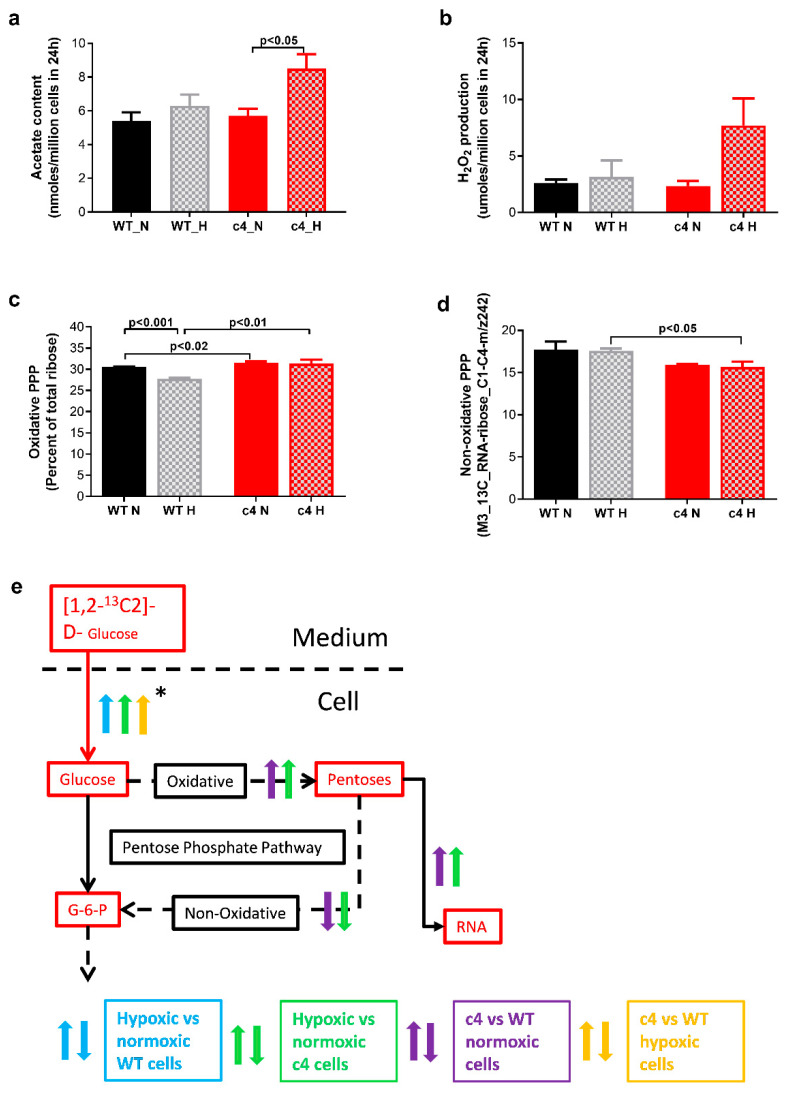
PFAO markers and differences in pentose phosphate pathway flux in Hepa-1 cells. (**a**) Acetate content significantly increased in Hepa-c4 cells in response to hypoxia (*p* < 0.05, *n* = 5). (**b**) Hypoxic Hepa-c4 cells released more hydrogen peroxide into the media (*p* < 0.05, *n* = 5) (**c**) Nucleic acid ribose synthesis via the oxidative pentose phosphate pathway was higher in Hepa-c4 than WT cells in both normoxia and hypoxia (*p* < 0.02 and *p* <0.01, respectively, *n* = 4). (**d**) Nucleic acid ribose re-assembly via the non-oxidative pentose phosphate pathway was slightly lower in Hepa-c4 than WT cells (*p* < 0.05), but was not substantially affected by hypoxia in either cell line. (**e**) Diagram showing flux differences and changes between the two Hepa-1 cell types in normoxia and hypoxia. N, normoxia; H, hypoxia (1% O_2_); arrows pointing upwards suggest significant increase for a given comparison. * Indicates differences in metabolic flux (↑ increased flux; **↓** decreased flux) due to HIF deficiency and hypoxia, which were inferred from changes in the relevant ^13^C-labelled isotopomers; *↑ and *↓ fluxes were inferred from concentration changes in the substances indicated, not their ^13^C-labelling. Panels (**a**–**e**) show data expressed as mean ± SEM; statistical significance assessed by unpaired two-tailed t-test.

**Figure 6 cells-11-03595-f006:**
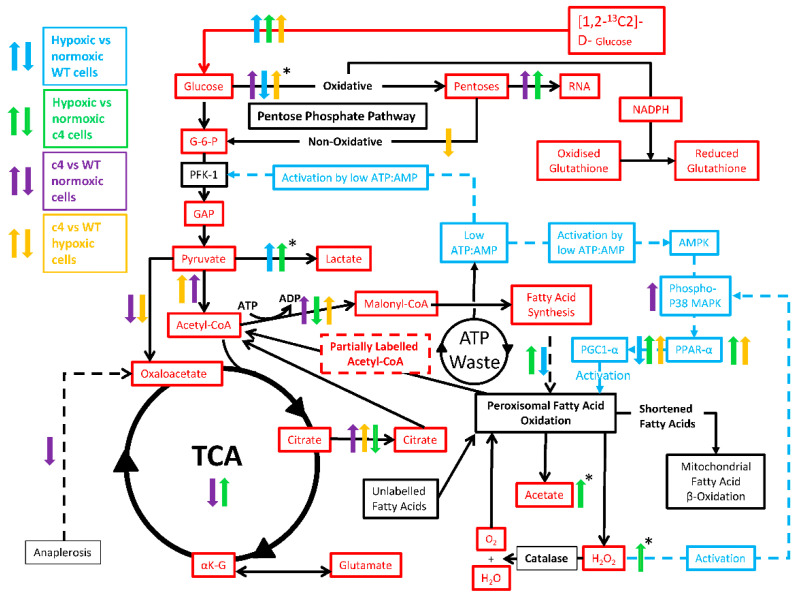
Metabolic fluxes and control factor expression changes in Hepa-1 cells. Solid black lines indicate metabolic pathways; dashed dark-blue lines indicate control mechanisms. Coloured up and down arrows indicate alterations in metabolic fluxes or changes in control factor expression compared with the corresponding parameter in WT or Hepa-c4 cells in normoxia: blue arrows, WT cells in hypoxia vs. normoxia; green arrows, Hepa-c4 cells in hypoxia vs. normoxia; purple arrows, Hepa-c4 cells vs. WT cells in normoxia; orange arrows, Hepa-c4 cells vs. WT cells in hypoxia. Differences in metabolic flux (↑ increased flux; **↓** decreased flux) due to HIF deficiency and hypoxia were inferred from changes in the relevant ^13^C-labelled isotopomers; *↑ and *↓ fluxes were inferred from concentration changes in the substances indicated, not their ^13^C-labelling. Arrows pointing upwards suggest significant increase for a given comparison.

## Data Availability

All data are available in the main text or Appendix A.

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
