# Peer review of "Survival Pathways of HIF-Deficient Tumour Cells: TCA Inhibition, Peroxisomal Fatty Acid Oxidation Activation and an AMPK-PGC-1α Hypoxia Sensor"

_cells, 2022, doi:10.3390/cells11223595_

Round 1

Reviewer 1 Report

The study shows a noteworthy aspect of tumour therapy by providing an interesting explanation to what looks like a paradox.

The research is simple indeed, but complete, well designed and conducted. 

However the manuscript, in my opinion, turns to be too long, thus Discussion/Conclusion should be shortened

Author Response

Thank you for reviewing our manuscript and your comments.

Comment 1: However the manuscript, in my opinion, turns to be too long, thus Discussion/Conclusion should be shortened.

Response: We have done some small editing to shorten the manuscript. However, we find it hard to shorten even more as it is complex and further clarifications were requested by 2 out of the 3 reviewers.

Reviewer 2 Report

This manuscript by Glinska et al, details experiments aimed at determining the survival pathways of HIF-deficient tumour cells. The authors evaluated some metabolic and molecular adaptations to hypoxia in a HIF-1β deficient Hepa-1c4, a hepatoma model lacking HIF1/2 signaling. They showed that these cells were capable of growing under hypoxia, and that glucose consumption and lactate production increased, accordingly with a previous study (ref #17). These cells exhibited higher glutamate, PDH, citrate shuttle and malonyl-CoA fluxes than normal Hepa-1 cells, whereas pyruvate carboxylase, TCA and anaplerotic fluxes decreased. Furthermore, hypoxic HIF-1β deficient Hepa-1c4 cells increased the expression of phospho-PGC-1α, phospho-p38 MAPK and PPARα, suggesting an activation of the AMPK pathway to survive under hypoxia. After having reported a  higher intracellular acetate, H2O2 secretion (suggesting increased peroxisomal fatty acid β-oxidation) and fatty acid synthesis and degradation, the authors suggested a further  contribution to upregulation of the AMPK pathway. They concluded that since these tumor cells can proliferate without the HIF-1/2 pathways, combinations of HIF1/2 inhibitors with PGC1 or AMPK inhibitors should be explored.

General comments.

This manuscript provides some data to further support the fact that the AMPK pathway is relevant for cell survival under hypoxia in the absence of HIF activation. In addition, the authors described a large variety of metabolites aimed to justify the metabolic and molecular adaptation  to survive  under hypoxia in the absence of HIF activation.

This study provides some very interesting and novel findings, obtained by using also elegant 13C tracer assays. However, some information regarding other pro-survival pathways should be added or, at least, discussed.

Major points:

1)      At least some information regarding the use of some AMPK inhibitors, in this cellular model under hypoxia, should be added (or, at least, discussed), as the authors mentioned in their conclusions.

2)      Besides AMPK, other pathways are relevant in the regulation of metabolic reprogramming in tumor cells. These include the ERK and JNK pathways (Papa et al. Oncogene 38, 2223–2240.2019). In addition, hypoxia induces phosphorylation of ERK1-2 and hypoxia-induced autophagy is accompanied by elevated Ras/ERK signalling with an essential role in the protection of cancer cells against hypoxic stress (Pursiheimo JP et al Oncogene 2009 28 334-344)

Is ERK1-2 phosphorylation differently affected in HIF-1β deficient Hepa-1c4 under hypoxia?

The overall information could be very useful to fully justify the title of this manuscript (“Survival pathways of HIF-deficient tumour cells…..)

Minor points:

1)      In the Materials and Methods section, lines 157-163 should be erased.

Author Response

Thank you for reviewing our manuscript and your comments. We found the advice constructive and have incorporated the suggestions into our revision:

Comment 1: At least some information regarding the use of some AMPK inhibitors, in this cellular model under hypoxia, should be added (or, at least, discussed), as the authors mentioned in their conclusions.

Response: We have addressed this comment in lines: 541-544

Specific and widely used inhibitors of AMPK are not yet available as recently reviewed in (Vara-Ciruelos D, Open Biol. 2019 Jul 26;9(7):190099). The most well-known inhibitor – compound C has been found non-specific (Liu X. Mol Cancer Ther. 2014 Mar;13(3):596-605). Other inhibitors have not yet been widely used.

Comment 2: Besides AMPK, other pathways are relevant in the regulation of metabolic reprogramming in tumor cells. These include the ERK and JNK pathways (Papa et al. Oncogene 38, 2223–2240.2019). In addition, hypoxia induces phosphorylation of ERK1-2 and hypoxia-induced autophagy is accompanied by elevated Ras/ERK signalling with an essential role in the protection of cancer cells against hypoxic stress (Pursiheimo JP et al Oncogene 2009 28 334-344). Is ERK1-2 phosphorylation differently affected in HIF-1β deficient Hepa-1c4 under hypoxia? The overall information could be very useful to fully justify the title of this manuscript (“Survival pathways of HIF-deficient tumour cells…..)

Response: We have addressed this comment in lines: 518 - 527 as advised.

Comment 3:  In the Materials and Methods section, lines 157-163 should be erased.

Response: Deleted accordingly – thank you.

Reviewer 3 Report

In the manuscript, the authors evaluated metabolic fluxes via the tricarboxylic acid cycle (TCA), citrate shuttle, fatty acid utilisation and pentose phosphate pathways using HIF-1β deficient hepatoma cell line. They revealed the hepatoma cell lines were able to grow in hypoxia despite the HIF-1β deficiency. Overall, the manuscript is well organized and clearly written. Therefore, I have only minor comments.

Major points.

The cells responded to hypoxia despite a deficiency of HIF-1β, but what other factors might be involved in the hypoxic response in this case should be discussed. It should also be discussed whether the present results would be similar if the cells were deficient in HIF-1α.

Minor points.

1.    The direction of the arrows (up or down) is confusing and needs to be clarified (e.g. Figure 2e, 3g, 4d, and 5e).

2.    Figure 4b: It would be preferable to show phosphorylated p38 by Western blotting, but not necessarily essential.

Author Response

Thank you for reviewing our manuscript and your comments. We found the advice constructive and have incorporated the suggestions into our revision:

Comment 1: Cells responded to hypoxia despite a deficiency of HIF-1β, but what other factors might be involved  the hypoxic response in this case should be discussed.

Response: We have addressed this comment in lines: 518 - 527 focusing especially on the ERK pathway. Also, lines 177-178 refer to a publication, where PGC-1a is proposed as a possible HIF-independent hypoxia sensor.

Comment 2: It should also be discussed whether the present results would be similar if the cells were deficient in HIF-1α.

Response: This response has been addressed in lines 535-538. We believe that the results may be different as with the inhibition of only HIF-1α, the functioning of HIF2 would not be perturbed.

Comment 3: The direction of the arrows (up or down) is confusing and needs to be clarified (e.g. Figure 2e, 3g, 4d, and 5e and also 6).

Response: All figures have now been amended to explain that arrows pointing upwards suggest a significant increase for a given comparison. For example, in 2e: blue, upward arrow shows significant increase in glucose uptake in hypoxic WT cells as compared to normoxic WT cells.

Comment 4: It would be preferable to show phosphorylated p38 by Western blotting, but not necessarily essential.

Response: Unfortunately, this is not available. The ELISA has high specificity and is quantitative, although we agree it would be desirable to have a blot.